# Technologies for Vitrification Based Cryopreservation

**DOI:** 10.3390/bioengineering10050508

**Published:** 2023-04-23

**Authors:** Mohammad Amini, James D. Benson

**Affiliations:** Department of Biology, University of Saskatchewan, Saskatoon, SK S7N 5E2, Canada

**Keywords:** cryobiology, physical chemistry, bioengineering, heat transfer, technology, vitrification

## Abstract

Cryopreservation is a unique and practical method to facilitate extended access to biological materials. Because of this, cryopreservation of cells, tissues, and organs is essential to modern medical science, including cancer cell therapy, tissue engineering, transplantation, reproductive technologies, and bio-banking. Among diverse cryopreservation methods, significant focus has been placed on vitrification due to low cost and reduced protocol time. However, several factors, including the intracellular ice formation that is suppressed in the conventional cryopreservation method, restrict the achievement of this method. To enhance the viability and functionality of biological samples after storage, a large number of cryoprotocols and cryodevices have been developed and studied. Recently, new technologies have been investigated by considering the physical and thermodynamic aspects of cryopreservation in heat and mass transfer. In this review, we first present an overview of the physiochemical aspects of freezing in cryopreservation. Secondly, we present and catalog classical and novel approaches that seek to capitalize on these physicochemical effects. We conclude with the perspective that interdisciplinary studies provide pieces of the cryopreservation puzzle to achieve sustainability in the biospecimen supply chain.

## 1. Introduction

Cryopreservation is one of the approaches to preserve cells and tissues over long time periods. Because of this, cryopreservation of cells and tissues is essential to modern medical science, including cancer cell therapy, tissue engineering, transplantation, reproductive technologies, and bio-banking. For example, one such application, liver cell transplantation (LCT), treats diseases involving liver metabolism [1,2,3,4]. In order to achieve sustainability in LCT, developing a liver cell bank of cryopreserved cells and tissues is critical to facilitate access to liver cells in clinical centers [5]. In 2017, 23.3% of all organ transplantations were liver transplants (32,348 transplanted livers) [6]. Although these numbers are high, the actual number of liver donors was much higher, i.e., 27% of donors’ livers were not transplanted due to limitations, resulting in many patients dying whilst on the waiting list. Early cancer therapy has improved the survival rates of women who need to cryopreserve their ovarian tissue due to gonadotoxic treatment [7]. Cryopreservation is also a vital part of the supply chain of new cell sources for regenerative medicine [8]. In response to climate change and global warming, a wide range of cells and tissues must be cryopreserved for both plant and animal species conservation [9,10,11]. Finally, rapid freezing methods have many advantages in histological applications compared to chemical preservation, such as chemical fixation applicable in electron microscopy [12].

Cryopreservation often results in significantly reduced cell and tissue viability due to ice crystal formation. This reduced viability is attributed to several classes of damage, including osmotically induced volume changes [13], damaging ice formation [6], cytosol concentration changes [14], and thermal stress due to cooling and warming [15]. Water solidifies as ice at low and moderate cooling rates and can form deleterious intra- and inter-cellular ice crystals [16]. During this gradual solidification, the extracellular media becomes divided into two parts. The first part is ice, generally considered pure water because ice cannot dissolve significant amounts of solutes. The second part is unfrozen water, which dissolves the remaining solutes. Therefore, the concentration of the solution increases gradually as the ice grows. This high solution concentration is considered to be a toxic cell environment [17].

Chemical and biological materials, called cryoprotectant agents (CPAs), have been crucial in decreasing damage due to ice formation and excessive ion concentrations. However, a high concentration of CPAs harms cells and tissues because it leads to cellular osmotic and mechanical stress as the result of volume changes (shrinking and swelling) during the addition and removal of CPA. In addition, biophysical and biochemical interactions between CPA and cytoplasmic constituents cause toxic effects in a biological sample [13,18,19,20,21,22,23].

One approach to avoid damage due to ice formation is to use ice-free cryopreservation, known as vitrification, to reduce the potential toxicity of high solute concentrations below 0 °C and avoid ice crystallization and growth. There is a non-crystalline state at extremely low temperatures in which cells and tissues are considered safe. This state is called a glassy or vitreous state, and the material forming in this state is called glass. A sufficiently concentrated solution can vitrify at any cooling rate [24], but these high concentrations are associated with significant toxicity. However, due to the formation of hydrogen bonds between water molecules, ice crystallization is a time-dependent phenomenon. Therefore, to avoid crystal formation and achieve a vitreous state under less toxic CPA concentrations, it is necessary to pass through the related states of heterogeneous and homogeneous ice formation as quickly as possible (Figure 1). The minimum cooling rate to vitrify pure water was estimated on the order of 10^8^ °C/min [25], not achievable for droplets with diameters more than 10 µm [26]. However, cells seemingly can tolerate ice crystals with dimensions less than 1 µm [27]. Hence, CPA-free cell cryopreservation has been carried out in droplets with a maximum volume of 200 pL, equivalent to a sphere of diameter 70 µm [26]. Therefore, it is generally desirable in “vitrification protocols” to cool and warm samples as rapidly as possible without causing damaging thermal stress.

To achieve rapid cooling rates there have been two primary methods to reach vitrification: moving a cryogen over the sample (convection method) and using solid surface cooling (conduction method) [28]. The convection method is generally limited by the Leidenfrost effect: when a biological sample directly contacts a cryogen such as liquid nitrogen, the large temperature difference creates a vapor film between the sample and the cryogen. This film acts as an insulating layer around the biological sample, resulting in reduced heat transfer coefficients and, thus, dramatically reduced cooling rates in the biological sample [29]. However, in solid surface cooling, the sample contacts a cooled surface with high thermal conductivity and diffusivity to maintain the temperature gradient. This method can avoid the Leidenfrost effect due to indirect contact between the sample and cryogen [28]. Another approach to reducing the required CPA concentration is to apply hydrostatic pressure because the glass transition temperature can be increased by hydrostatic pressure [30]. In addition, with increasing hydrostatic pressure, the melting point of water reduces [31,32,33]. Therefore, high hydrostatic pressure can reduce the temperature difference between the melting point and the glass transition temperature. In short, with or without the increase of pressure, the goal of vitrification in each of these methods is to pass through this temperature range at a high cooling rate, preventing large ice crystals formation or growth (Figure 1) [34].

Each approach is associated with significant hurdles, including contamination due to direct contact between biospecimen and liquid nitrogen [35] and the Leidenfrost effect [30]. Moreover, most solid surface devices cool a sample from just one side. Consequently, cooling depth reduces, and satisfactory survival depth is not achievable [16]. Therefore, the usual tradeoff is that immersion in a cryogen removes heat from all sides but encounters reduced cooling rates, and solid surface cooling removes heat faster, but only from one side. An apparatus that can use solid surfaces to cool a contained biospecimen from two sides and reduce CPA concentration by applying a high cooling rate could partially address this dilemma [16].

In this review, we discuss the physical chemistry of cryopreservation and, specifically, vitrification. Further, we explore the effect of hydrostatic pressure in vitrification and on biological samples. Finally, a number of methods, technologies, and devices used in vitrification are presented, and results are summarized.

## 2. Physical Chemistry Aspects of Vitrification

### 2.1. Vitrification

As pure liquid water is cooled below its melting temperature (*T*_m_), it first passes into a region called the supercooled liquid state, as shown in Figure 1 [36]. In this region, water molecules combine to form a cluster due to Brownian motion as a base for ice nuclei formation [17]. Based on classical nucleation theory [37], water below melting temperature is in a metastable state corresponding to a minimum of the appropriate thermodynamic potential without any crystal formation [38]. If, by stochastic fluctuations, an ice-aggregate of a sufficiently large size is formed, known as critical cluster, it can further grow in accordance with thermodynamic evolution laws. The radius of the critical cluster has quite different values depending on pressure and temperature. It tends to infinity, near the melting temperature. At the radius of the critical cluster, more water molecules favor thermodynamically binding the critical cluster and making it larger. Afterward, this critical cluster may enable the entire supercooled liquid to experience a significant phase transition, known as freezing.

The number of critical clusters per unit of volume in a specific temperature can be calculated thermodynamically. Ice critical cluster formation in the aqueous solution necessitates an entropy decrease and maybe creates a barrier against nucleation. The energy barrier is the result of the related change in Gibbs energy to create the critical cluster containing a number of water molecules. Therefore, the energy barrier must be conquered to form the ice critical cluster. The energy barrier includes a volume term and a surface term. The volume term is related to the chemical potential decrease of the critical cluster with respect to the supercooled state due to phase change. The surface term is related to the energy required to create the critical clusters interface.

Although liquid water can be prolonged in the supercooled state, homogeneous nucleation may occur at any temperature below melting temperature (*T*_m_) with a highly different probability. Homogeneous nucleation temperature (*T*_h_) is where homogeneous nucleation becomes increasingly probable [39]. A maximum nucleation rate exists depending on the temperature, occurring near the conventional glass transition temperature (*T*_g_), at which a glassy or vitreous state of non-crystalline material can form [40]. This transition is not associated with a specific temperature where the vitreous state forms but rather a temperature interval at which the process takes place. On the other hand, water impurities and container surfaces can contribute to heterogeneous ice nucleation earlier in supercooled water due to the energy barrier deduction of heterogeneous nucleation compared to homogeneous nucleation [41].

However, if the cooling rate is sufficiently high, these ice nuclei do not have sufficient time to grow due to water molecule diffusion limitations. By including a cryoprotectant agent (CPA), the medium viscosity increases [42], facilitating an ice-free medium, or at least a medium with only tiny ice crystals, known as the “non-harmful” ice crystals, which can be achieved in the supercooled region and below *T*_m_ without substantial ice crystal growth. This kind of ice crystal is too small to be seen by an electron microscope [28]. On the other hand, because of an insufficient cooling rate due to the thickness of the biological sample or the volume of the medium, “harmful” ice crystals can form and be seen by the electron microscope since there is enough time and energy for water molecules to develop ice crystal nuclei [43]. To sum up, after cryoprotectant toxicity, the most influential factor that influences cell and tissue viability is ice crystal size.

In glasses, the values of thermodynamic response variables, such as the specific heat and coefficient of expansion, are closer to those of the crystalline form than to the liquid state [40] (Figure 2). In contrast, x-ray diffraction studies show that the structure of glasses, such as silicate glasses [44,45], resembles the structure of their liquid form. In fact, the enthalpy, entropy, and specific volume of the glass are the same as those for liquids, which is why there is no latent heat in the transition of glass formation [40]. On the other hand, the rate of change of these thermodynamic state variables is close to those of the crystalline form. This is mainly because glass is a form of liquid with high viscosity. In a method reported in the literature, Boutron [46,47,48,49] and others [50,51,52,53] measure the glass transition temperature (*T*_g_) based on a sudden change in heat capacity using a differential scanning calorimeter (DSC) [24].

### 2.2. High Hydrostatic Pressure

Due to the influence of high hydrostatic pressure on reducing the melting temperature and increasing the glass transition temperature, the application of high hydrostatic pressure towards preserving biological systems commenced in the 1970s to reduce ice damage during freezing [54]. This technique has been widely used for biospecimen fixation with applications in electron microscopy [55,56]. However, there are few studies of cryopreservation of living cells and tissues under hydrostatic pressure and ultra-rapid cooling, and further studies are required to investigate living specimens during cryopreservation [57].

Under ambient pressure, water freezes in a hexagonal form known as Ice I [31]. As shown in Figure 3, by increasing the hydrostatic pressure, the melting temperature of water reduces. This process continues until about 205 MPa (2050 bar), where Ice II or Ice III form high-pressure shapes of ice. After this point, with increasing pressure, the melting temperature of high-density configurations of ice increases. Therefore, the minimum melting temperature of the water is −23 °C (250 K) under 205 MPa pressure. Moreover, pressure can influence water viscosity, resulting in an increase in the glass transition temperature [30]. In this way, and similar to the concentration of solutes, decreasing the melting temperature and increasing the glass transition temperature, the high hydrostatic pressure may improve vitrification.

To establish a theoretical foundation for further experimental designs, Miyata et al. investigated the vitrification of aqueous glycerol solutions under high pressures at low cooling rates (3 °C/min) [30]. Miyata et al. observed that the temperature corresponding to the maximum homogeneous nucleation rate decreased with increasing pressure up to about 200 MPa, and after that, this temperature increased with pressure. In contrast, the glass transition temperature (*T*_g_) increased linearly with pressure, but the slope decreased as solution concentration decreased and would be zero for pure water.

At atmospheric pressure, water is less viscous. Therefore, as the ice grows in the medium, additional water molecules are recruited to make larger ice crystals. However, under high pressure, water becomes much more viscous. Consequently, the growth rate of ice crystals is reduced by high pressure. At atmospheric pressure, freezing causes ice crystals with rough surfaces, resulting in mechanical injury to the membrane of cells and tissues [58]. On the other hand, the surface of ice crystals at high pressure is much smoother and could possibly cause less mechanical damage to the cell membrane [59]. One advantage of applying high pressures in cryopreservation is the potential to reduce or eliminate ice crystal formation in much deeper regions inside a biological sample [19]. In cryomicroscopy, where the preservation of ultrastructure is paramount and post-thaw viability is not a priority, Sartori et al. found they could increase the vitrification depth more than 10-fold by applying high pressure during freezing [60]. Heuser and Staehelin [28] claimed that it would be possible to vitrify up to the depth of 600 μm in planar samples and 1 mm in spherical samples by high pressure in comparison with the 40 μm depths that was achieved at atmospheric pressure.

### 2.3. Warming

Recrystallization is defined as the growth of ice crystals in the warming process [24]. Ice nucleation happens rapidly at lower temperatures due to the low free energy barrier. In fact, the vitrified solution can transform into ice crystals at temperatures higher than *T*_g_ due to the metastability of the solution with respect to ice formation. This nucleation forms a base for extensive ice crystal growth at higher solution temperatures. For example, the peak of the ice nucleation rate occurs around *T*_g_, and ice nucleation even happens at temperatures lower than *T*_g_. This is because in order to form ice nuclei, local replacement of molecules is all that is needed [61]. However, ice growth occurs at higher temperatures. At lower temperatures, the solution is more viscous, which limits the diffusion of water molecules to the crystal clusters. On the other hand, in dilute solutions, there are regions in which ice nucleation and growth overlap with a broader range [62] because water molecules share hydrogen bonds with each other rather than with solute molecules and are, therefore, able to participate in ice crystal growth at lower temperatures. In this regard, the warming rate that significantly prohibits ice crystal formation and growth during warming is called the “critical warming rate” [46]. Tiny ice crystals are highly likely to be formed in ultra-rapid freezing. However, cells can seemingly survive the presence of tiny ice crystals [63,64,65,66]. This is where the warming rate can play a significant role. At slow warming rates, the vitrified solution recrystallizes and forms large ice crystals, damaging cells [63,64]. For example, Mazur et al. [67] cryopreserved Chinese hamster cells by fast freezing coupled with ultrafast warming—clearly, adding heat energy is simpler than removing it. These cells had higher survival compared with those cryopreserved at the slow freezing rate. The authors claimed that one reason could be the lower salt concentration due to lower cell dehydration associated with rapid freezing. However, by slow warming, de Graaf and Koster [68] reported low viability in rapidly frozen rat liver slices. By DSC technique, Amini and Benson [69] show that, under ultra-rapid freezing, ice recrystallization does not happen in low CPA concentrations during slow warming; however, in moderate and high CPA concentrations, ice recrystallization is a dominant part of ice formation in slow warming.

In some references, the temperature related to crystallization in the warming process is called recrystallization temperature (*T*_r_) [28,42,70]. In a solution with high molecular weight solutes, *T*_r_ is recorded at higher temperatures. This *T*_r_ has practical implications: Bank [71] rapidly warmed yeast cells from liquid nitrogen and at −50 °C in the duration of 24 hr did not observe any ice crystal damage under the condition of CPA-free. However, Bank reported visible ice crystals at −45 °C after 30 min and at −20 °C after just 5 min. There was massive damage inside the cells.

Rall et al. [72] mentioned that *T*_r_ is −50 °C for mouse embryos. Gilkey and Staehelin [28] did not observe any recrystallization in the range of −95 and −50 °C in a large number of samples and under CPA-free conditions. In the case of CPA-free vitrification, it seems that *T*_r_ is reduced. For example, a CPA-free suspension of red blood cells has *T*_r_ of −30 °C, while *T*_r_ is −80 °C for this suspension with 30% glycerol [73].

## 3. Biological Aspects of Cryopreservation

### 3.1. Cryopreservation of Isolated Cells and Tissue Slices by the Method of Vitrification

Cryopreservation facilitates the nearly indefinite storage of cells and tissues in a state of biological arrest [74]. The temperature of storage is critical: proteases are still active in the range from −20 °C to −80 °C, and between −80 °C and −130 °C, ice crystal formation is most likely due to the high homogeneous nucleation rates. Thus, the storage of biological samples is limited at these temperatures, but below −130 °C, there is no usable thermal energy for biochemical reactions [5]. Because of the massive shift in thermodynamic properties at the glass transition temperature and because the minimal *T*_g_ of water/CPA systems is around −140 °C, this is considered the maximal safe temperature for long-term storage of cryopreserved cells and tissues [75,76,77,78].

Generally, a cell and tissue “bank” is a collection of cryopreserved cells and tissues [68] that can be warmed as the needs arise. In fact, Luyet performed the first vitrification in 1937 [79], and the first success in live human birth by vitrified sperm was in 1953 [80]. Vitrification of liver slices (human) was attempted by Wishnies et al. in a highly concentrated solution of 4.7 mol/L 1,2-propanediol [81]. They placed nylon mesh containing slices into the liquid nitrogen directly with an estimated cooling rate of 5000 °C/min, and after warming, reached the high inherent biotransformation rate compared to fresh slices.

Different approaches have improved the vitrification of precision-cut liver slices (PCLS). For example, de Kanter and Koster [82] showed that by stepwise CPA addition (where the final concentration is reached after intermediate steps), they were able to improve the viability of rat liver slices when they used rapid freezing (250 °C/min) and 12% DMSO. Similarly, de Kanter et al. [83] claimed that human liver slices cryopreserved successfully by rapid freezing had 66% viability compared with fresh slices. de Graaf et al. [84] found that increasing DMSO concentration to 18% in pre-incubation before freezing could enhance some viability metrics.

Another approach for improving rapid freezing in PCLS was using a high thermal conductivity material such as aluminum plates in the freezing and thawing process [85]. In this method, the samples are placed between aluminum plates separated by a gasket and submerged in liquid nitrogen. This is why a higher cooling rate in the cryopreservation of liver tissues is crucial compared to isolated rat hepatocytes [86,87,88,89]. This can be explained by the difference between the structure of isolated cells and tissue slices as a group of cells together. In isolated cells, the extracellular ice crystals do not damage the cells; however, ice crystals accumulated between the cells in tissue slices can injure the cell and tissue structure [90,91]. In contrast to slow freezing, extracellular ice crystals do not form significantly during successfully implemented rapid freezing.

The cytoplasm tends to vitrify more readily than the extracellular medium because of the high concentration of intracellular proteins [92,93]. Furthermore, due to the structure of cells and tissue, water is not homogenously distributed and is divided into smaller portions. This water discontinuity in the intracellular and intercellular spaces can postpone the formation of ice crystals because, by decreasing the volume, the probability of ice formation reduces [94,95]. Therefore, lower concentrations of CPA are required to achieve vitrification inside cells and tissues compared to the extracellular/extra-tissue solutions. According to these concepts, de Kanter and Koster [82] postulated that during rapid freezing, water vitrifies inside the slices and forms ice crystals outside the slices, and they suggest that cells in the inner layer of slices can be protected by outer layers exposed to ice crystals formed in the medium.

Several sample-containing devices, or cryocarriers, designed to facilitate vitrification have been developed, including cryoloop, straw (and many variations thereof), Cryotip, Cryopette, VitriSafe, S3 μS-VTF, S3 system, Rapid-i [96,97,98,99]. These cryocarriers have seen considerable success but have a number of limitations. To wit, because the cooling rate is dependent on the sample volume, the cooling rate is reduced, especially in the interior part of the sample, due to thermal diffusivity, and this necessitates higher CPA concentrations to avoid ice crystals at the center of the tissue and decreases the viability of the sample [100,101,102]. Because of this, most vitrification cryocarriers are tailored to very small volume samples (<0.1 µL), which means applications to all but the smallest tissues (e.g., no larger than embryos) are precluded. Additionally, while these commercially available cryocarriers achieve high heat transfer, they are still usually limited by the Leidenfrost effect, which reduces the heat transfer between the sample and liquid nitrogen [29,103,104]. Finally, one pressing issue in using open vitrification carriers, which do not entirely cover the biospecimen, is contamination due to the sample’s direct contact with liquid nitrogen associated with disease transfer to and from the samples [105,106,107,108,109,110,111,112,113,114,115,116,117,118].

Table 1 presents a number of cryopreserved samples, cryocarriers, and outcomes using the vitrification method.

### 3.2. Effect of Hydrostatic Pressure on Biological Samples

Pressure, similar to temperature, is a thermodynamic parameter that can affect any thermodynamic system, such as biomolecular media. In nature, the range of pressure that can be tolerated by living species ranges from 0.1 MPa (~1 atm) as an atmospheric pressure to 110 MPa (~1100 atm) at the deepest point of Mariana Trench at a depth of 11 km in the western Pacific Ocean [143].

According to the principle of Le Chatelier, if an equilibrated system is exposed to an alteration, the equilibrium is adjusted to object to the alteration [144]. For example, a pressure increase leads to a change in the equilibrium of the system toward its smallest volume during a constant temperature process [145,146]. In addition, the compressibility of biological materials, made of mostly water, is very low. For instance, the compressibility of erythrocytes is about 4 × 10^−10^ Pa^−1^ [147]. Therefore, by increasing the pressure to a cryobiologically relevant amount, say to 100 MPa, the cellular volume change would be approximately 4%.

The use of pressure as an additional control variable for vitrification is not without its own hazards. In the literature, the effects of high hydrostatic pressure on biological samples are divided into two categories: “physiological high hydrostatic pressure” (pHHP), which is between 1 and 100 MPa, and “non-physiological high hydrostatic pressure” (HHP), which is above 100 MPa [148,149]. Marsland and Zimmerman investigated the biological effects of physiological high hydrostatic pressure (pHHP). They stated that pHHP dissolved the mitotic spindle apparatus and stopped chromosome movement in the eggs of Arbacia punctulata [150,151,152]. After removing pressure, these effects disappeared [153,154]. Haskin et al. found pressure influences the cytoskeleton of osteosarcoma cells and their responses, which were similar to those under a heat shock [155,156]. Wilson et al. showed morphology and cytoskeleton changes under pHHP [157,158]. Lammi et al. reported the influences of pHHP on articular tissue and confirmed that these effects were reversible and had no impact on the viability of the cells [149]. In terms of pressure tolerance in organs, dog kidneys tolerated 100 MPa for 20 min [159], while rabbit kidneys under 50 MPa were injured after 20 min [160].

The induction of an external shock, such as heat and pressure, was suggested as a mechanism for initiating stress induction [161]. When cells are subjected to stresses such as temperature increase or decrease, heat shock proteins (HSP) are upregulated to respond to these stresses. HSPs not only act as a protection to inhibit severe injuries to cells but also play a role as signals to induce immune responses. Categorized by molecular weight, there are different HSPs in mammalian cells: HSP20, HSP60, HSP70, HSP90, and HSP100 [162]. In the case of imposing pressure of 30 MPa over chondrocytes, only HSP70 was detected [163,164,165]. Experiments have shown that short-term pHHP on articular cartilage and chondrocytes did not significantly affect cell viability [161]. However, long-term pHHP had different results. For example, Islam et al. reported apoptosis in chondrocytes exposed to 5 MPa for 4 h [166].

When pressure greater than 100 MPa is applied, changes in protein structure and even protein denaturation [167,168,169,170,171,172], enzyme alteration and dysfunction [173,174,175,176], and changes in phospholipid bilayers from fluid-crystalline to a gel-like form [170,172,177,178] were reported. On the one hand, several investigators found induction of apoptosis in cells exposed to 100 MPa pressure, such as murine erythroleukemia cells [179,180] and human lymphoblasts [179]. On the other hand, other researchers did not find a significant effect of 100 MPa pressure on human cell viability [181,182,183]. As a result, it is hypothesized that the cell cycle plays a role in pressure-induced apoptosis [179,184].

Moving to higher pressures, Frey et al. [161] found that at 200 MPa, cell death occurred through apoptosis identified by symptoms such as DNA fragmentation and morphological changes and that pressurizing to more than 300 MPa has been shown to result in necrosis because of plasma membrane damage [185,186,187]. In addition, an increase in cytoplasm viscosity was observed [161]. In fact, the rate of increase and duration of pressure substantially affect cell death. Although a fast non-physiological pressure (HHP) rate (the change in pressure over time) was associated with significant cell death, a short holding time of HHP had a minor impact on cells [161].

### 3.3. Cytotoxicity of Cryoprotectants in Vitrification

Cryoprotective agents (CPAs) are defined as chemicals that mitigate or eliminate ice damage and reduce high ionic concentrations during cryopreservation [188]. As evidenced by the phase diagram in Figure 1, it is possible to cool cells and tissues without any ice formation with an unlimited amount and concentration of CPAs. In practice, there are two kinds of CPA. The first is “penetrating” or “permeating” CPAs, which have low molecular mass and can pass through the cell membrane and include (among others): ethylene glycol, propylene glycol, dimethylsulfoxide, glycerol, formamide, and methanol. The second is “non-permeating” CPAs with high molecular mass and cannot pass cell membranes, such as trehalose, sucrose, polyethylene glycol, polyvinyl alcohol, and polyvinyl pyrrolidone [189,190]. In fact, CPAs cause changes in hydrogen bonds between water molecules to inhibit ice formation [191]. While the CPA concentration increases, the translational movement of water and CPA molecules reduces [192,193,194]. The reduction of the water molecule diffusion coefficient postpones the gathering of water molecules in supercooled or heterogeneous regions to form ice crystal clusters, resulting in delaying ice formation (melting point decreases) and promoting glass formation [190]. The translational, vibrational, and rotational movements of water molecules, restricted in the vitreous state, are reduced in aqueous solutions with CPA [195,196]. As a result, the glass transition temperature increases in solutions containing CPAs.

However, concentration-dependent CPA toxicity has been identified as a limitation in cryopreservation [197]. Especially in vitrification, this issue is the main obstacle to reaching the glassy state without any ice formation [198]. Cells and tissues react differently when exposed to different CPAs [199], and most of these differences are related to the conditions of the experiment, including CPA type and concentration, temperature, and exposure time. In this section, we explore the concepts of toxicity of CPA to reduce the amount of toxicity and achieve acceptable cryopreservation.

CPA toxicity has been frequently reported in the field of cryobiology. One of the toxicities related to CPAs is cell membrane toxicity. At the outer and inner surfaces of cell membrane bilayers, there are hydrophilic polar head groups, while in the middle of the cell membrane, there are hydrophobic fatty acid chains. Some factors can increase the permeability of CPAs, such as lipophilicity. On the other hand, increasing molecular size and hydrogen bond formation can reduce a CPA’s permeability [200]. For example, in molecular dynamics (MD) simulations, it was shown that sugar molecules interfaced with the lipid head group, while there was no considerable change in the lipid bilayer of cell membranes [201,202,203]. In cell membranes, CPA molecules can replace water molecules in their hydrogen bonds with phospholipids. For example, trehalose, sucrose, maltose (up to 2 mol/kg), and glucose (up to 4 mol/kg) have been shown to substitute 20–25% of water-phospholipid hydrogen bonds with CPA–phospholipid hydrogen bonds [203]. In addition to the influence on cell membranes, DMSO has another effect called pore formation, occurring in high concentrations of DMSO [204,205,206,207,208]. In this regard, three ranges of concentration can be described. In the first range, the DMSO solution has a relatively low molar concentration (2.5–7.5 mol %) associated with thinning the lipid bilayer membrane by lateral expansion. There is not any bond between DMSO and lipid head groups. Therefore, DMSO molecules stand close to phosphate groups inside the lipid molecules [207]. In the second range of DMSO molar concentration (10–20 mol %), penetration of DMSO molecules inside the lipid bilayer molecules becomes intensive, and DMSO molecules can make transient water pores through cell membranes. Finally, increasing the DMSO molar concentration to more than 25 mol % results in collapsing of the lipid bilayers [207,208].

Warner et al. [209] investigated the toxicity kinetics of multi-CPA mixtures by a toxicity cost function approach and compared the results with experimental data. Their model in their cell type showed that ethylene glycol and glycerol are relatively non-toxic CPAs compared to formamide, which is a highly toxic CPA. Moreover, by increasing the type of CPAs in the solution, the total toxicity of the solution reduces. Benson et al. [210] have provided a similar model able to quantify the CPA toxicity and optimize equilibration protocols in tissues. Their model shows that time-optimal protocols can perform faster than the non-optimized standard stepwise method but incur more accumulated toxicity. Benson et al. [211] defined the minimization of the toxicity cost function to optimize protocols for CPA addition and removal, while they believed that CPA does not need to be included in the extracellular solution during CPA removal. In addition, they stated that osmotic and mechanical damage due to volume change is insignificant when cell volume is placed within the cell volume tolerance limit.

To sum up, many factors play a role in the CPA toxicity of biospecimens, requiring specific cryoprotocol to reduce cytotoxicity based on cryopreserved biospecimen, CPA type, and freezing method.

## 4. Bioengineering Aspects of Vitrification

### 4.1. Necessity of Ultra-Rapid Cooling

The growth of ice crystals can cause severe damage to cells and tissues. Thus, to address this issue, the use of ultra-rapid cooling is needed when low CPA concentrations are used. Ultra-rapid freezing is a cryopreservation technique much faster than slow freezing and even conventional vitrification methods in cryobiology. Slow freezing methods often require expensive programmable freezers with complicated protocols to cryopreserve each type of cell and tissue. On the other hand, the success of any fast-freezing approach is greatly dependent on the CPA concentration, which is generally associated with toxicity in cells and tissues. In short, vitrification approaches will trade the challenge of managing subzero ice formation and water transport via cooling rates in slow or equilibrium protocols for the challenge of managing CPA toxicity and mechanical stresses at above-zero temperatures in vitrification protocols.

### 4.2. Methods and Devices of Vitrification

Cooling rates are generally limited by conductive heat transfer in tissues greater than 1 mm thick. A number of methods and devices have been presented to maximize heat transfer, and these methods fall into three categories used for rapid freezing: conventional rapid freezing, ultra-rapid freezing, and high-pressure freezing [28,36].

#### 4.2.1. Conventional Rapid Freezing

In the conventional rapid freezing method, a biological sample is immersed directly in a cryogen, such as liquid nitrogen, with a boiling point of −196 °C. In this method, the heat loss from the sample creates an insulating film of evaporated nitrogen around the sample, called the Leidenfrost effect. Therefore, heat loss and cooling rate are reduced [19]. However, this method has been widely used in research and clinical centers due to its simplicity and low cost.

#### 4.2.2. Ultra-Rapid Freezing

Concerning ultra-rapid freezing technology, there are four methods: plunging, spraying, jetting, and metal block [19].

##### Plunge Freezing

Plunge freezing commenced in the 1950′s in the application of electron microscopy [212]. Among the method of ultra-rapid freezing, this method is less costly, more straightforward, and most used. Plunging involves placing a biological sample either with or without a container directly in liquid cryogens [71,213,214,215,216,217,218,219,220,221,222,223,224,225,226,227]; however, compared with the conventional rapid freezing method, there are four effective conditions to improve the cooling rate: High entry velocity of the sample into a cryogen (minimum 1 m/s) [228,229,230,231]; High surface-to-volume ratio of the sample (e.g., a copper mesh or a device such as a cryoloop or cryotop) [120,121,122,125,216,217,222,232,233]; The lowest possible temperature of cryogen (e.g., nitrogen slush or liquid helium) [232,234,235,236,237,238,239]; Stirring the cryogen in a container to decrease temperature gradients [235,240].

Due to a number of improvements, this approach has a higher cooling rate than conventional rapid freezing; nonetheless, it faces significant challenges in freezing thick biospecimens, such as 1 mm thick tissue.

##### Spray Freezing

The spray freezing method is the same as plunging, but a sample is reduced to small droplets in order to increase the surface-to-volume ratio of contact area [28]. In this regard, Akiyama et al. [26] developed an inkjet system to create small droplets containing cells. Droplets were placed on a surface cooled by liquid nitrogen. Cooling droplets with a solid surface could prevent the Leidenfrost effect. The volume of droplets was between 40 and 200 pL (10^−12^ L), they achieved a maximum cooling rate of 2.2 × 10^6^ °C/min, and they confirmed the vitrification of droplets by spectroscopy and a high-speed camera. Although a high cooling rate is achievable with this method for CPA-free cryopreservation, this system is limited in scope to cells in suspension, and as such cannot be used for tissue slices or other small tissues.

##### Jet Freezing

In the jetting method, liquid cryogen is shot toward a sample at a high velocity [241,242,243,244]. Katkov et al. [29] designed a device that could shoot a jet of liquid nitrogen to a solid surface containing a sample. The whole cooling process was performed in milliseconds or even microseconds, and they could not measure the sample temperature directly because of the thermal inertia of the thermocouple. Thus, they used glycerol solution with a known critical cooling rate as a function of the concentration and identified vitrification of the medium visually when the medium was completely transparent. They claimed a cooling rate of 10^5^ °C/min for a plastic cryovial containing 4 mL 15% glycerol. However, this cooling rate was not measured directly and, as such, may be an overestimation of the cooling rate. This approach may have other problems, such as not eliminating the Leidenfrost effect, completely.

##### Metal Block Freezing

In metal block freezing, a sample is placed into direct contact with a highly heat-conductive metal block cooled to as low a temperature as possible [232,245,246,247]. Cold metal block cooling was tried by Simpson in 1941 [248]. After that, several researchers, principally in the field of cryo-electron microscopy, improved this technique (Van Harreveld and Crowell [249], Van Harreveld et al. [250], Heuser et al. [246,247], Boyne [245], Escaig [232], Philips and Boyne [251], Heath [252]). In all these devices, a metal block made of copper or silver is cooled by liquid nitrogen or helium and maintained in an insulating container. By passing cooled nitrogen or helium gas over the surface of the metal block, heat transfer limiting frost formation can be prevented. A sample is placed in a carrier mounted in a holder assembly. Then, this holder falls onto the metal block through guidance, and the rebound of the sample after striking the block is prevented by a spring. This technique was used first for freezing tissues for cryo-electron microscopy, and the results were acceptable (Heuser et al. [246], Hirokawa and Kirino [253], Ichikawa et al. [254], Ornberg and Reese [255], Hirokawa and Heuser [256], Terracio et al. [257], Hirokawa [258], Schnapp and Reese [259], McGuire and Twietmeyer [260], Philips and Boyne [251], Menco [261]). This approach was also developed to freeze cellular suspensions (Chandler and Heuser [262,263], Ornberg and Reese [264], Chandler [265], Wagner and Andrews [266]). This method has an estimated cooling rate of 10^6^ °C/min, but only for the surface (Plattner and Bachmann [267]). However, due to cooling just from one side, the depth of vitrification is limited and may not be sufficient to vitrify the entire sample. Moreover, this technique can damage the biological sample because of the impact onto the cooled surface. In addition, these biological samples were bare without any closed container, which becomes a challenge in clinical applications.

There have been many designs, modifications, developments, and alterations based on the idea of a cold metal block [268,269,270,271,272]. However, reports of these approaches beyond cryo-electron microscopy and into labs or clinical centers have been rare, even though these relatively inexpensive approaches yield good results. This may be due to the complicated designs that make results difficult to reproduce.

#### 4.2.3. High-Pressure Freezing

In high-pressure freezing technology, in one method, a sample is preserved by high-pressure jetting liquid nitrogen [29,60,273,274,275,276,277]. The main issue regarding this method is the Leidenfrost effect due to direct contact between liquid nitrogen and biological sample and damage of the sample because of the direct liquid nitrogen jet. In another method, isochoric (constant volume) can cause high pressure inside the sample due to the increasing ice volume during freezing [278,279,280]. However, this method of high-pressure freezing cannot be categorized as a vitrification method because of the low cooling rate used in this process [32].

A summary of the freezing techniques, samples tested, corresponding advantages, disadvantages, and achievements is given in Table 2.

## 5. Summary

This review investigates the physiochemical concepts involved in cryobiology, various cryopreserved biospecimens, and devices designed to improve biospecimens’ viability after thawing. Up to now, a few manuscripts have had an interdisciplinary viewpoint of all factors playing a role in cryopreservation. Here we study several significant parameters influencing the vitrification technology from different aspects.

Recently, vitrification has been considered widely in stem cells, cell therapy, reproductive technology, and transplantation. Many protocols, methods, devices, and technologies have been used and tested to improve vitrification. However, there are impediments to vitrification technology, which can be solved by considering all disciplines involved in cryopreservation, such as physical chemistry, biomathematics, engineering, and biology.

CPA toxicity is a significant concern that should be tackled through an interdisciplinary approach, including physical chemistry techniques such as differential scanning calorimetry (DSC) and mechanical and mathematical modeling of transport and toxicity [208,210]. It is essential to define low-CPA concentration cryoprotocols or even CPA-free methods to reduce cytotoxicity and cell mechanical damage.

The idea of “non-harmful” ice crystals in cryobiology can enhance cell and tissue cryopreservation [27]. This idea requires ultra-rapid cooling with techniques that can increase heat transfer. High heat transfer in the ultra-rapid method is limited not only by the thermal properties of devices but also by the biospecimen thickness and cryosolution volume, imposing substantial restrictions against this technique. In this regard, many devices and methods have been introduced and developed to address this matter.

The warming process is another concern in cryobiology. Cryosolutions with high CPA concentrations, due to the high amount of vitrification in the solution during cooling, ice recrystallization is a significant obstacle to cell and tissue viability in the warming process. Under these circumstances, a high warming rate is required to suppress devitrification and large ice crystal formation [305]. In fact, there are a number of engineering approaches to increase warming rates in ever larger tissues, including laser, microwave, and induction-based warming [306,307,308,309]. However, in cryosolutions with low CPA concentrations under ultra-rapid cooling, it is plausible that ice recrystallization could be eliminated at a lower warming rate when only tiny ice crystals form in the cooling process [69].

To sum up, there are several limitations for the vitrification method in practical applications, such as low biospecimen volume, contamination, small tissue thickness, complicated techniques, and devices that need to be addressed through interdisciplinary investigations. There is no doubt that cryopreserved specimens will be highly demanded in the future, particularly in cancer cell therapy for cancer and on-shelf organ cryopreservation for transplantation [310].

## 6. Conclusions

In this review, we presented some basic and interdisciplinary notions applicable to cryopreservation and a diverse cryopreserved biospecimen with different methods and devices, showing various results due to the type of cells and tissues with different cooling rates and cryoprotocols. Hence, it is essential to pay more attention to fundamental interdisciplinary concepts of cryobiology to build a bridge between all current gaps in the literature and achieve sustainability in this realm of technology.

## Figures and Tables

**Figure 1 bioengineering-10-00508-f001:**
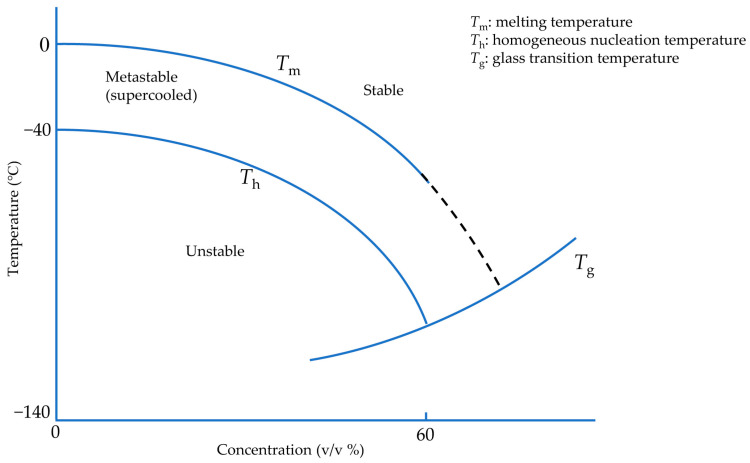
Idealized and modified phase diagram of CPA-water solution during vitrification. Adapted with permission from Ref. [24], 2023, Elsevier.

**Figure 2 bioengineering-10-00508-f002:**
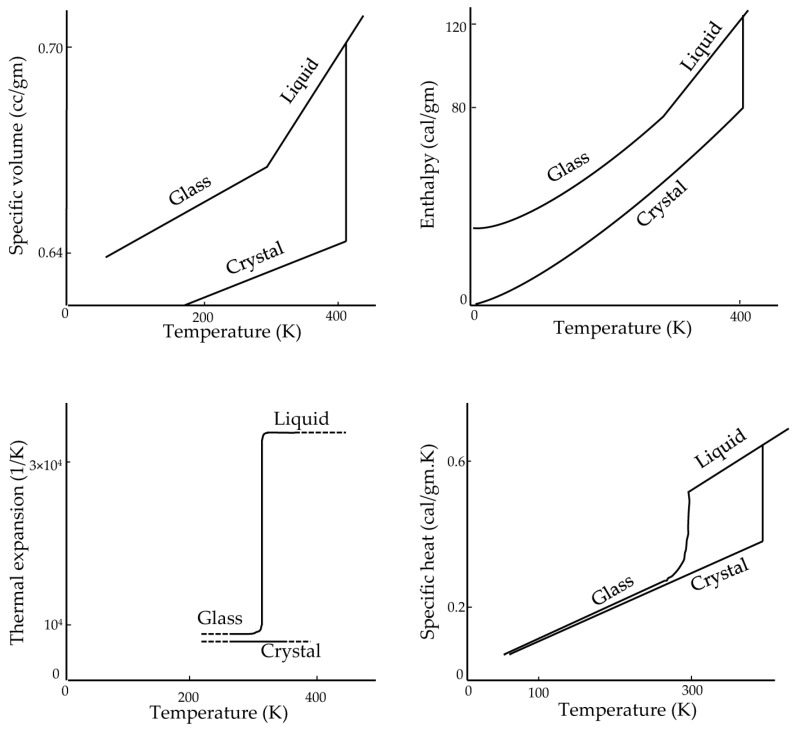
Thermodynamic properties of glucose. Adapted with permission from Ref. [40], 2023, American Chemical Society.

**Figure 3 bioengineering-10-00508-f003:**
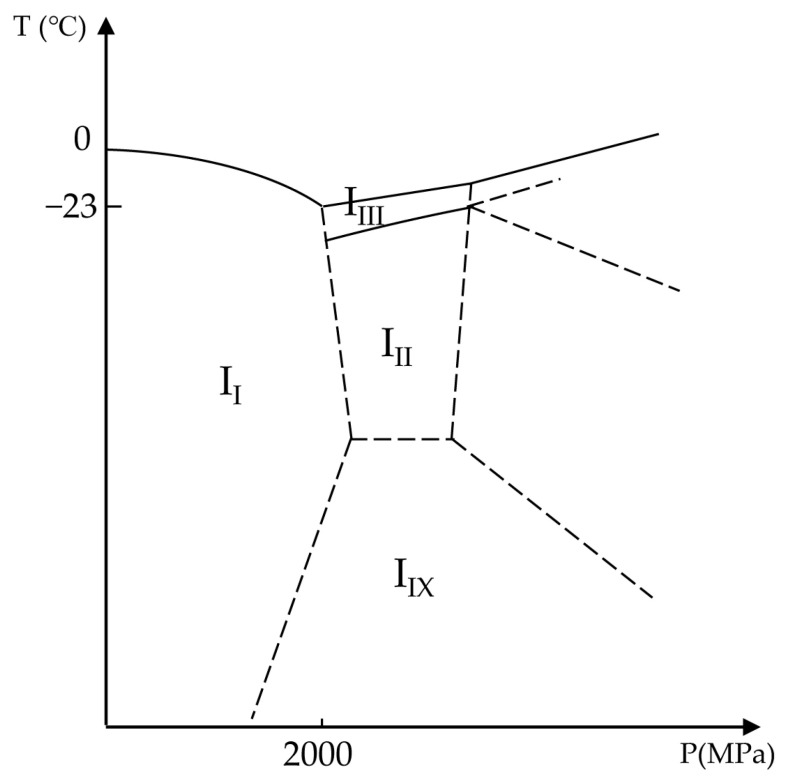
Phase diagram of ice as a function of temperature and pressure. I_I_, I_II_, I_III_, and I_IX_ are ice shapes of Ice I, Ice II, Ice III, and Ice IX, respectively. Adapted with permission from Ref. [31], 2023, Elsevier.

**Table 1 bioengineering-10-00508-t001:** Cell and tissue slice cryopreservation by vitrification.

Sample	CPA Solution	Cryocarrier	Sample Size	Viability Evaluation	Outcome	Ref.
Ovine embryo	15% EG + 15% DMSO + 0.5 M sucrose + 30% Ficoll 70	Cryotop-Spatula	<0.1 µL	Embryo morphology Cell membrane integrity with propidium iodide	79.7% viability	[119]
Cow blastocyst	16% EG + 16% DMSO + 0.5 M sucrose	Fork	0.5 µl	Immunostaining	74% hatching rate with warming in straw	[120]
Donkey embryo	15% EG + 15% DMSO + 0.5 M sucrose + 18% Ficoll 70	Cryotop	<1 µL	Transrectal ultrasonography to track follicular activity and confirm ovulation Cell membrane integrity with propidium iodide and Hoechst 33342	70% viability	[121]
Bovine oocytes	2% EG + 2% PG	Cryotop	<1 µL	Oocyte viability, nuclear maturation status, embryo development, and blastocyst quality	15% blastocyst yield	[122]
Mouse embryo	20% EG + 0.4 M sucrose + 24% Ficoll 70	Cryotube	5 μL drop	Developmental ability	98.7% viability	[123]
Human ovarian tissue	30% EG + 0.5% trehalose + 6% FBS	Cold solid-surface, Straw, drop size into LN_2_	100 µL	Histologic analysis	Higher morphological follicles by the method of drop size into LN_2_	[124]
Ovine cumulus-oocyte complexes	VS1, VS2	Plastic insemination straw	250 µL	Morphological evaluation, Trypan blue staining	54.5% viability, 16.8% ultrastructural changes	[125]
Bovine oocyte	15% EG + 15% DMSO + 1.0 M sucrose	Hollow fiber	Inner diameter 200 μm, wall thickness 15 μm	Acetolacmoid staining	23% blastocyst yield	[126]
Human embryo	15% EG + 15% DMSO + 0.5 M sucrose	Plastic blade	Thickness: 0.05 mm	Assessment of pregnancy by transvaginal ultrasound imaging	100% viability in Blastocyst	[127]
Bovine oocytes	7.5% EG + 7.5% DMSO	Silk fibroin sheet multilayer	0.1 mm thickness, 0.7 mm width, 10 mm depth	Morphological survival rates	23% blastocyst yield	[128]
Rabbit chondrocyte sheets	20% EG + 20% DMSO + 0.5 M sucrose + 10% COOH-PLL	Sealable polyethylene bag and nylon meshes	110 × 85 mm; film thickness: 0.063–0.064 mm	Trypan blue staining Histological examination Immunohistochemical staining	91% viability	[129]
Human liver tissue	4.7 M 1,2-propanediol	Honeycomb-like tray	Diameter: 1 cm, Thickness: 200–250 µm	Xenobiotics metabolism	7-EC metabolism, 7-HC conjugation (Results are very unstable)	[81]
Human ovarian tissue	7.5% EG + 7.5% DMSO + 20% FBS + 13.5% EG + 13.5% DMSO + 0.5 mol/l sucrose	Needle directly into LN_2_	1 mm^3^	Histologic Analysis by H&E Ultrastructural evaluation using TEM TUNEL assay for detection of apoptosis Assessment of tissue damage using an LDH assay	Higher viability in stroma cells and lower apoptotic primordial follicles	[130]
Human ovarian tissue	10% DMSO + 10% EG	Direct cover vitrification (DCV)	1 × 1 × 1 mm	Follicle examination using electron microscopy and TUNEL	Higher normal follicles and lower apoptotic cells	[131]
Ovine testicular tissues	18% EG + 18% DMSO + 0.5 M trehalose	E. Vit (modified plastic straw)	1 mm^3^	Cell plasma membrane integrity	73.6% viability	[132]
Human ovarian tissue	2.62 mol/L DMSO + 2.6 mol/L acetamide + 1.31 mol/L PROH + 0.0075 mol/L PEG	20 μL droplet into LN_2_	4 mm × 4 mm × 1.5 mm	Immunohistochemistry histology	Higher follicles growth	[133]
Cat testicular tissues	15% EG + 20% glycerol + 0.5 M sucrose	Needle	1–2 mm^3^	Seminiferous tubule Morphology Mitochondrial activity Cell composition	92.9% viability	[134]
Human ovarian tissue	40% EG + 1 M sucrose + 30% ficoll 70	Cryovial	2 mm^3^	Histological examination Molecular assessment Hormonal assay Immunocytochemistry	95.5% viability	[135]
Mouse testicular tissues	15% EG + 15% DMSO + 0.5 M sucrose	Metal grid	Tissue fragments (0.5–1 mm^2^)	Trypan blue staining Hematoxylin and eosin (H&E) staining Immunohistochemistry staining	97.7% viability	[136]
Dog ovarian tissue	15% EG + 7.5% DMSO + 0.5 M sucrose + 2.5% PVP	Needle	Diameter: 2 mm	Neutral red staining Histology Xenotransplantation assays	94.5% follicular viability	[137]
Rat testicular tissues	15% EG + 15% DMSO + 0.5 M sucrose	Inoculation loop	Pieces of approximately 3 mm	Trypan blue staining histological evaluation	84.8% viability	[138]
Rabbit trachea	18% EG + 22% DMSO + 0.5 M sucrose	Cryotube	0.5 cm × 0.5 cm	Morphological and ultrastructural assessment HE examination TUNEL assays TEM and SEM	97% viability	[139]
Rat kidney tissue	VM3 (8.44 M) in VS4 buffer (7.5 M)	Cryovial	Diameter: 5 mm	ATP content Histological integrity	Cortex: histomorphology (86%), ATP content (113%); medulla: histomorphology (79%), ATP content (68%)	[140]
Rat liver tissue	VM3 (8.44 M) in VS4 buffer (7.5 M)	Cryovial	Diameter: 8 mm	ATP content Histological integrity	histomorphology (71%), ATP content (58%)	[140]
Human osteochondral dowel	9.5% EG + 18% DMSO + 5.8% PG + 14.1% glycerol + 0.1 mg/mL chondroitin sulfate (CS)	5 mL vial	Full thickness in 10 mm diameter	Membrane integrity Metabolic activity Histology Immunohistochemistry	75.4% viability	[141]
Pig osteochondral dowel	16.6% EG + 21.5% DMSO + 22% PG + 0.1 mg/mL CS	Conical tube	two diameter sizes (10.0 mm and 6.9 mm) with 10 mm thickness	Chondrocyte assessment with membrane integrity stain and the chondrocyte metabolic activity by Alamar Blue.	60–80% viability	[142]

VM3 and VS4 are commercial CPA, consisting of buffer components such as NaCl, NaHCO_3_, KCl, …, and CPAs such as DMSO, Formamide, and Ethylene glycol.

**Table 2 bioengineering-10-00508-t002:** Different freezing techniques to enhance the cooling rate, suppress ice crystal formation, and reach vitrified biosolution for various applications, including cryomicroscopy and cryopreservation.

Technique	Advantages	Disadvantages	Sample	Result	Ref.
Conventional rapid freezing	-Simple -Less costly -Most widely used	-Low cooling rate -Leidenfrost effect -High CPA concentration	Bovine endothelial cells TE-85 strain human osteosarcoma cells Human breast carcinoma cells	Freeze-fracturing of cells in monolayers or multilayer tissue cultures	[281]
			Human spermatozoa	No difference in motility was detected compared with slow freezing	[282]
			Human spermatozoa	No difference in DNA fragmentation was detected compared with slow freezing	[283]
			Rat ovarian tissue	Follicular pool reduced.	[284]
			Mouse follicle	No significant differences in meiotic spindle formation	[285]
			Human ovarian tissue	Similarity of genes expression in vitrified and non-vitrified groups	[286]
Plunge freezing	-Relative simplicity, low cost, most widely used	-Sophisticated compared to the conventional technique	Human erythrocytes	Two opposing 25–30 μm surface zones were frozen in the apparent absence of ice	[228]
	-Higher cooling rate compared to the conventional method	-Low cooling rate for thick biospecimen	Rat liver tissue	Increasing the cooling rate	[232]
			Maize root epidermal cells	Achieving ice crystals being acceptably small and non-disruptive	[287]
			PtK2 cells	Reducing shrinkage and alteration of the trabecular structure	[217]
			Fibroblasts or epithelial cells derived from Xenopus laevis embryos	Observation of the cytoplasm included interconnected filaments. The general organization was similar to intact cells.	[288]
			Hyphal tip cells of Fusarium acuminatum	Observing all cellular membranes and most organelles due to the transparency of freezing	[222]
			Whole eyes of adult albino mice	The low depth freezing due to the thickness of samples	[289]
			Slices of rat kidney	Reaching high-quality morphological preservation	[235]
			Antennae of the silk moth, Bombyx mori	Observation of freezing damage just in deeper tissue regions under CPA-free and probably due to natural CPA	[215]
			Bovine oocytes and blastocysts	8% blastocyst yield	[290]
			Murine and bovine oocytes and embryos	73% cleavage rate 7% blastocyst rate	[291]
			Cattle oocytes	55.81% cleavage rate 11.24% blastocyst rate	[292]
Spray freezing	-High cooling rate due to the high surface-to-volume ratio of biospecimen	-Limited to particulates up to 1 µm in diameter -Costly and tedious	5% glycerol-water solution (cryomedia sample without biospecimen)	Preventing large ice compartment A smooth surface of frozen water	[293]
		-Biospecimen damage due to shearing forces in the spray nozzle	Paramecium tetraurelia 7S (wild type) cells	No Ca^+ 2^ influx, which is necessary for induction of membrane fusion.	[294]
			Rat hepatocytes	Higher viability and better morphology in the bulk droplet (3–5 mm diameter) compared to controlled rate freezing	[295]
			3T3 mouse fibroblast cells, human neuroprogenitor cells (NPCs)	90% Viability with CPA concentration less than 0.8 M in 114 nL droplet	[296]
Jet freezing	-Quite high cooling rate due to the high velocity of the cryogen jet -Fast process	-Difficulty in sample handling -Biospecimen damage due to direct jetting with high velocity	Bovine chromaffin cell	Increasing rapid cooling rate up to 40,000 °C/s	[297]
	-High depth of cooling due to cooling from both sides of the specimen at atmospheric pressure	-Non-uniform freezing through the biospecimen -Biospecimen adhesion to cryodevice holder after freezing	Yeast cells	Enhancing the cooling rate to 18,000 °C/s	[243]
	-Moderate cost of equipment and operation	-Less advantage in tissue samples compared to the cold metal technique	Spinal cord explants from mouse embryos	Retaining the shape of the tissue surface	[242]
			Rat Sertoli cells Human monocytes	High cooling rate even in the absence of any CPA	[298]
			Human embryonic stem cells, human spermatozoa	Higher cell membrane integrity compared to conventional freezing	[299]
			15% glycerol-water solution (cryosolution sample without biospecimen)	Vitrification of 4000 µL of cryosolution	[29,300]
Metal block freezing	-High thermal diffusivity of copper and silver block	-The high cost and challenging nature of manufacturing and operating copper and silver blocks due to their softness	Red blood cells	Modifying ice microcrystals appeared at a depth of 25–30 μm, while it was already observed at a depth of 12–15 μm.	[232]
		-Biospecimen damage due to crushing, shearing, and bounce	Mouse liver tissue	Obtaining acceptable blocks of preserved tissues for cryofixation	[249]
		-Low depth of cooling due to cooling from just one side -Bare biospecimen without container causing contamination	Mouse liver and diaphragm tissues, Isolated strands of celery phloem, rose phloem, pea root tip, and Chinese cabbage and tobacco leaf tissue	Using the heat-conductive properties of copper to increase the cooling rate The tissue surface layer was ice-free up to 12 μm in depth.	[269]
			Goat testicular cell	74.8% cell viability	[301]
			Leydig cells (murine cell line TM3)	Superior cell growth, mitochondrial activity, and cytoplasmic esterase enzyme activity than the conventional method	[302]
			Mouse myoblast C2C12 cells, rat primary mesenchymal stem cells	More than 80% cell viability in 40 pL CPA-free droplet	[26]
			Cat ovarian tissue	Normal follicles after thawing	[303]
			Cow blastocyst	69% hatching rate	[304]
High Pressure (hyperbaric) freezing	-Higher depth of cooling due to cooling from both sides of the specimen under high pressure -Lower CPA concentration -Ability to cool ultra-rapidly thick biospecimen such as tissue	-Biospecimen preservation quality for cryomicroscopy is not as good as well-prepared biospecimen under ultra-rapidly frozen at atmospheric pressure.	Rat brain tissue	Preventing large ice crystal formation in tissue up to 0.5 mm thick	[274]
	-Lower rate of ice crystal formation and growth	-The least known and least tested technique -High temperature and low cooling rate for the isochoric condition	Beef liver catalase crystals	Increasing the depth of vitrification approximately ten times larger with freezing at high pressure (2 Kbar)	[60]
			Ascites tumor cell	Obtaining the same quality of vitrification without CPA	[273]
			Rat cartilage tissue	Enhancing the quality of preserved chondrocytes than those preserved by chemical fixation	[275]
			Bovine and rat parathyroid cell	Excellent preservation of ultrastructure of cells	[276]
			Actomyosin system of Physarum polycephalum	Avoiding artificial alterations observed in chemical fixation	[277]
			Rat heart	The first freezing of mammalian organs without CPA at −4 °C, 41 MPa, and under the isochoric condition	[278]
			Fish muscle tissue	No cellular dehydration and maintaining the morphology of the frozen tissue under the isochoric condition	[279]
			Madin-Darby canine Kidney epithelial cells (MDCK)	60% and 18% cell viability during 60 and 120 min at –10 °C under the isochoric condition	[280]
			Human red blood cells	8% or less hemolysis of RBCs with 5% DMSO (*v/v*) or 8% glycerol (*v/v*) and 120 MPa pressure	[58]

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
