# Peer review of "Technologies for Vitrification Based Cryopreservation"

_bioengineering, 2023, doi:10.3390/bioengineering10050508_

Round 1
Reviewer 1 Report
The paper is well-motivated and I started to read it initially with great interest. However, I have a set of critical minor and some more even principal comments concerning the treatment of crystal nucleation and vitrification by the authors.
First, the caption of Fig.1 reads „Idealized and modified phase diagram of CPAwater solution during non-equilibrium vitrification [24]“. Vitrification means the freezing-in of a metastable liquid into a non-equilibrium state but what is meant by „non-equilibrium“ vitrification? The second of the curves specified with Th is not explained neither in the caption nor in the text when the figure is mentioned for the first time. This is confusing. This bug is removed later but anyway.
In Section 2.1 it is stated incorrectly: „A liquid below melting temperature is out of thermodynamic equilibrium but can still remain locally (heterogeneously) metastable without any crystal formation where it can have minimum local free energy [37]“. Instead, a liquid below the melting temperature is not out of thermodynamic equilibrium but in a metastable state corresponding to a minimum of the appropriate thermodynamic potential. The notations „locally (heterogeneously) metastable“ is incorrect and „it can have minimum local free energy“ are senseless. Senseless is also the reference to the paper by Kauzmann in this connection. It is a great paper but the origin of metastability was described much earlier by J. W. Gibbs (1876-78). I do not like also the formulation „Ice crystals need a base to grow and become large enough“, it is misleading. Instead, if by stochastic fluctuations an ice-aggregate of a sufficiently large size is formed (denoted as critical cluster) it can further grow in accordance with thermodynamic evolution laws. And the radius of the critical cluster (not the „critical radius“) has quite different values in dependence on pressure and temperature. Near to the melting temperature, it is not equal to 10-9 m („The minimum size of this cluster of water molecules has a critical radius of 10-9 m [38]–[42]“) but tends to infinity. Incorrect are also the statements „The free energy in the process of ice nucleation is divided into two parts: volume free energy and surface free energy [28]“, „The volume free energy is considered negative due to the higher number of water molecules inside the cluster than the number of water molecules outside the cluster“, „the surface free energy is positive due to fewer interactions between water molecules at the surface of cluster and others in the medium“, „If the net free energy is zero or negative, the cluster can become an ice crystal nucleus, achieving a “critical radius”“ and so on. The authors should consult some elementary textbooks on nucleation theory to correct this part. This refers also to the next paragraph
where the authors explain what they understand under the notation Th.
Homogeneous nucleation may occur at any temperature below Tm but with
highly different probabability. There exists a maximum of the nucleation rate in dependence on temperature and this maximum is located near to the conventional glass transition temperature, so the statement „At extremely low temperatures, there is a transition at which a glassy or vitreous state of noncrystalline material can form [37]“ has also to be corrected. By the way, a bit later the authors acknowledge this fact as well. The last paragraph in section 2.1 contains also a number of bugs, it has to be reformulated.
With respect to Section 2.2, I have to note that the origin of the increase of Tg with change of pressure consists in the pressure induced increase of viscosity and not as stated „pressure can influence the degrees of freedom of water molecules, resulting in increasing the glass transition temperature [30]“. The increase of viscosity with increasing pressure similarly explains the decrease of the temperature corresponding to the maximum of the homogeneous nucleation rate with increase of pressure. I repeat, the notation Th discussed in this connection again is senseless.
The first paragraph of Section 3.1 has to be reformulated for similar reasons as mentioned above.
The principle of le Chatelier is different as compared to the statement
„According to the principle of Le Chatelier, pressure increase leads to a change in the equilibrium of the system toward its smallest volume [134],[135]“. And once pressure can be changed at constant temperature, wo what is meant by „Temperature change due to pressure alteration in aqueous solutions and most biological materials is small“?
I have no queries concerning the outline of the different methods how
vitrification is realized. So, these parts seem to me to be fine.
Some minor bugs as I suppose:
1. Cryopreservation is one of the only approaches
2. liver transplants. (32,348 transplanted
3. Warming -> Heating
4. Pressure, like temperature, is a thermodynamic parameter that can affect
bimolecular media-> Pressure affects any thermodynamic systems.
5. In nature, the range of pressures supported by life -> What does this
mean?!
Summarizing, I consider the review as interesting but the serious flaws
mentioned have to be removed prior to publication. I recommend major
revision.
Author Response
(Note: Line numbers refer to those in the "Marked Changes" Manuscript.
The paper is well-motivated and I started to read it initially with great interest. However, I have a set of critical minor and some more even principal comments concerning the treatment of crystal nucleation and vitrification by the authors.
1) a. First, the caption of Fig.1 reads „Idealized and modified phase diagram of CPA water solution during non-equilibrium vitrification [24]“. Vitrification means the freezing-in of a metastable liquid into a non-equilibrium state but what is meant by „non-equilibrium “vitrification?
AU: “Equilibrium vitrification” can refer to protocols that use high enough concentrations of CPAs that they are never below T_h during cooling and warming, and “non-equilibrium vitrification” is the opposite. However, this classification at this point in the manuscript is not helpful, so we have deleted it in L80.
- The second of the curves specified with Th is not explained neither in the caption nor in the text when the figure is mentioned for the first time. This is confusing. This bug is removed later but anyway.
AU: We have included a more comprehensive legend for Figure 1.
2) a. In Section 2.1 it is stated incorrectly: „A liquid below melting temperature is out of thermodynamic equilibrium but can still remain locally (heterogeneously) metastable without any crystal formation where it can have minimum local free energy [37]“. Instead, a liquid below the melting temperature is not out of thermodynamic equilibrium but in a metastable state corresponding to a minimum of the appropriate thermodynamic potential. The notations „locally (heterogeneously) metastable“ is incorrect. and „it can have minimum local free energy“ are senseless.
AU: Thank you for this comment. We were not careful in our original text. We have revised this text entirely section 2.1. Vitrification.
- Senseless is also the reference to the paper by Kauzmann in this connection. It is a great paper but the origin of metastability was described much earlier by J. W. Gibbs (1876-78). I do not like also the formulation
AU: We have added the classic Gibbs reference. [38] Gibbs, A. :; Willard, J. On the Equilibrium of Heterogeneous Substances. Transactions of the Connecticut Academy of Arts and Sciences 1879, 3, 108–248.
- „Ice crystals need a base to grow and become large enough“, it is misleading. Instead, if by stochastic fluctuations an ice-aggregate of a sufficiently large size is formed (denoted as critical cluster) it can further grow in accordance with thermodynamic evolution laws.
AU: We have revised this as part of our rewrite of section 2.1. Vitrification.
- And the radius of the critical cluster (not the „critical radius“) has quite different values in dependence on pressure and temperature. Near to the melting temperature, it is not equal to 10-9 m („The minimum size of this cluster of water molecules has a critical radius of 10-9 m [38]–[42]“) but tends to infinity.
AU: We have revised this as part of our rewrite of section 2.1. Vitrification.
- Incorrect are also the statements„The free energy in the process of ice nucleation is divided into two parts: volume free energy and surface free energy [28]“, „The volume free energy is considered negative due to the higher number of water molecules inside the cluster than the number of water molecules outside the cluster“, „the surface free energy is positive due to fewer interactions between water molecules at the surface of cluster and others in the medium“, „If the net free energy is zero or negative, the cluster can become an ice crystal nucleus, achieving a “critical radius”“ and so on. The authors should consult some elementary textbooks on nucleation theory to correct this part.
AU: We have revised this based on classical nucleation theory as part of our rewrite of section 2.1. Vitrification. (in L133-142).
- This refers also to the next paragraph where the authors explain what they understand under the notation Th. Homogeneous nucleation may occur at any temperature below Tm but with highly different probability. There exists a maximum of the nucleation rate in dependence on temperature and this maximum is located near to the conventional glass transition temperature, so the statement „At extremely low temperatures, there is a transition at which a glassy or vitreous state of noncrystalline material can form [37]“ has also to be corrected. By the way, a bit later the authors acknowledge this fact as well.
AU: We have revised this as part of our rewrite of section 2.1. Vitrification.
- The last paragraph in section 2.1 contains also a number of bugs, it has to be reformulated.
AU: We have carefully copyedited this part as part of our rewrite of section 2.1. Vitrification.
3) a. With respect to Section 2.2, I have to note that the origin of the increase of Tg with change of pressure consists in the pressure induced increase of viscosity and not as stated „pressure can influence the degrees of freedom of water molecules, resulting in increasing the glass transition temperature [30]“.
AU: We have changed the text in L 232 to:
“Moreover, pressure can influence water viscosity, resulting in an increase in the glass transition temperature [30].”
- The increase of viscosity with increasing pressure similarly explains the decrease of the temperature corresponding to the maximum of the homogeneous nucleation rate with increase of pressure. I repeat, the notation Th discussed in this connection again is senseless.
AU: We have explained this in the manuscript based on previous worthwhile comments in L242-244:
“Miyata et al. observed that the temperature corresponding to the maximum of homogeneous nucleation rate decreased with increasing pressure up to about 200 MPa, and after that, this temperature increases with pressure.”
and L248-253:
“At atmospheric pressure, water is less viscous. Therefore, as the ice grows in the medium, additional water molecules are recruited to make larger ice crystals. However, under high pressure, water becomes much more viscous.”
4) The first paragraph of Section 3.1 has to be reformulated for similar reasons as mentioned above.
AU: According to previous comments, we have corrected the first paragraph of Section 3.1 in L312-313:
“The temperature of storage is critical: proteases are still active in the range of -20 ℃ to -80 ℃, and between -80 ℃ and -130 ℃, ice crystal formation is most likely due to high homogeneous nucleation rates.”
5) a. The principle of le Chatelier is different as compared to the statement „According to the principle of Le Chatelier, pressure increase leads to a change in the equilibrium of the system toward its smallest volume [134],[135]“.
AU: We have rewritten the section referring to “The principle of Le Chatelier” based on a new reference and illustrated with an example in L381-386:
“According to the principle of Le Chatelier, if an equilibrated system is exposed to an alteration, the equilibrium is adjusted to object to the alteration [140]. For example,pressure increase leads to a change in the equilibrium of the system toward its smallest volume during a constant temperature process [141],[142].”
- And once pressure can be changed at constant temperature, so what is meant by „Temperature change due to pressure alteration in aqueous solutions and most biological materials is small“?
AU: Our sentence was not relevant to this paragraph, and we have deleted in L384-386.
6) I have no queries concerning the outline of the different methods how vitrification is realized. So, these parts seem to me to be fine.
7) Some minor bugs as I suppose:
- Cryopreservation is one of the onlyapproaches
AU: We have considered your suggestion and deleted “only” in L24.
- liver transplants.(32,348 transplanted
AU: We have revised this in L31.
- Warming -> Heating
AU: “Warming” and “thawing,” instead of “heating,” are very common in cryobiology.
- Pressure, like temperature, is a thermodynamic parameter that can affect
bimolecularmedia-> Pressure affects any thermodynamic systems.
AU: We have corrected this misspelling in L377.
- In nature, the range of pressures supported by life -> What does this
mean?!
AU: We have rewritten this sentence as “In nature, the range of pressure that can be tolerated by living species …” in L378.
8) Summarizing, I consider the review as interesting but the serious flaws mentioned have to be removed prior to publication. I recommend major revision.
Reviewer 2 Report
In general, this review is disorganized and doesn’t provide new presentation of the most up-to-date data of the topic. The review cites very old references with a few references of the last 3 years.
Specific comments:
L17-18: correct “physicochemical effects …”
L38-40: This statement is irrelevant to the context of the work.
L40-41: Give examples for the chemical preservation. Is it for live cells/tissues?
Figure 1: The legend should illustrate all the abbreviations in the figure. Is there any copyright issue?
Figure 2 lacks X and Y axes details.
Numbering and subheading are inconsistent:
(3.) is focused on the biological aspects of cryopreservation, while 3.2 is the effect of hydrostatic pressure on biological samples, and 3.3. is the toxicity.
(4.) bioengineering with only a single sub number (4.1)!
This section is the core of the manuscript and should be illustrated in details and comprehensive manner, with exposing the advantages and disadvantages of each method.
In Table 2, spray method, what is meant by sample of 5% glycerol? Also what is the differences between the erythrocytes and red blood corpuscles?
(5.) discussion of what? The review should present the most recent data regarding the topic and discuss the concepts, applications, pros and cons of these data.
Author Response
(Note: Line numbers listed here correspond to the "Marked Changes" version of the manuscript.)
In general, this review is disorganized and doesn’t provide new presentation of the most up-to-date data of the topic. The review cites very old references with a few references of the last 3 years.
AU: We would like to express our gratitude to reviewer #2 for providing insightful and helpful comments. Our research focuses on the application of ultra-rapid cooling and high pressure techniques in cryopreservation, building on the foundations laid by earlier researchers in the 1980s who first explored these methods for the purpose of cryomicroscopy. To better reflect this context, we have added and referenced several recent studies on fast cooling in Table 2 and L535, L537. This will provide a more substantial basis for our work and demonstrate the advancements in the field.
1)L17-18: correct “physicochemical effects …”
AU: We have corrected this expression in L17-18.
2) L38-40: This statement is irrelevant to the context of the work.
AU: Although we meant to mention the general application of biomaterial freezing at low temperatures such as storage and delivery of mRNA vaccines, we have valued your comment and deleted this application in L38-40.
3) L40-41: a. Give examples for the chemical preservation.
AU: Chemical preservation has been common in electron microscopy, and according to your valued suggestion, we have added this example in L40-43.
- Is it for live cells/tissues?
AU: As stated in reference [1], it is not feasible to examine living cells and tissues due to the vacuum environment in electron microscopes. Instead, they must undergo a fixing process such as chemical fixation and rapid freezing methods, including high pressure freezing.
- Decelle, J.; Veronesi, G.; Gallet, B.; Stryhanyuk, H.; Benettoni, P.; Schmidt, M.; Tucoulou, R.; Passarelli, M.; Bohic, S.; Clode, P.; Musat, N. Subcellular Chemical Imaging: New Avenues in Cell Biology. Trends in Cell Biology 2020, 30 (3), 173–188. https://doi.org/10.1016/J.TCB.2019.12.007.
4) Figure 1: The legend should illustrate all the abbreviations in the figure. Is there any copyright issue?
AU: We have included a more comprehensive legend for Figure 1.
We will defer to the journal editor to assess whether our reinterpretation of these figures is subject to any copyright issues.
5) Figure 2 lacks X and Y axes details.
AU: We have added details on X and Y axes in Figure 2.
6) Numbering and subheading are inconsistent:
- (3.) is focused on the biological aspects of cryopreservation, while 3.2 is the effect of hydrostatic pressure on biological samples, and 3.3. is the toxicity.
AU: We tried to separate the physicochemical discipline (section 2) from the biological discipline (section 3). For example, we reviewed high hydrostatic pressure from the physicochemical viewpoint in section 2.2, while the biological effect of hydrostatic pressure has been discussed in section 3.2 from the biology viewpoint.
In section 3.3, we intended to review the biological influences of CPA on biospecimens from the viewpoint of biology. Therefore, we have changed this subheading to “Cytotoxicity of cryoprotectants in vitrification.”, revised in L433.
- (4.) bioengineering with only a single sub number (4.1)!
AU: This useful comment has been addressed in section 4. We have categorized section 4 to “4.1. Necessity of ultra-rapid cooling” and “4.2. Methods and devices of vitrification”. In addition, we have defined subheadings for 4.2 including: “4.2.1. Conventional rapid freezing”, “4.2.2. Ultra-rapid freezing”, and “4.2.3. High pressure freezing”. Due to various methods in 4.2.2, we have subcategorized this subsection to “4.2.2.1. Plunge freezing”, “4.2.2.2. Spray freezing”, “4.2.2.3. Jet freezing”, and “4.2.2.4.Metal block freezing”.
7) This section [4] is the core of the manuscript and should be illustrated in details and comprehensive manner, with exposing the advantages and disadvantages of each method.
AU: We agree and have now outlined the advantages and disadvantages of each method have been in Table 2 and added text in section 4 to discuss each method.
8) a. In Table 2, spray method, what is meant by sample of 5% glycerol?
AU: In this study, a 5% glycerol-water solution, a cryomedia sample without biospecimen, was used. To avoid any confusion, we have added this explanation in Table 2.
- Also, what is the differences between the erythrocytes and red blood corpuscles?
AU: In this study, it is mentioned that red blood corpuscles were used as the biosample. According to the reference [2], both are the same. However, we prefer to use red blood cells and have made changes throughout our manuscript to be consistent.
- Shemona, J. S.; Chellappan, A. K. Segmentation Techniques for Early Cancer Detection in Red Blood Cells with Deep Learning-Based Classifier—a Comparative Approach. IET Image Processing 2020, 14 (9), 1726–1732. https://doi.org/10.1049/IET-IPR.2019.1067.
9) (5.) discussion of what? The review should present the most recent data regarding the topic and discuss the concepts, applications, pros and cons of these data.
AU: We intended to summarize and have a conclusion on obstacles in vitrification methods and how to address these impediments in future research. To illuminate our purpose, we have changed the subheading to “Summary” in L621.
Round 2
Reviewer 1 Report
As I noted in my first review, „The paper is well-motivated and I started to read it initially with great interest. However, I have a set of critical minor and some more even principal comments concerning the treatment of crystal nucleation and vitrification by the authors.“ This situation is actually not changed although a number of corrections have been introduced inot the revised version. For example, people dealing with crystallization have found that in a variety of materials the temperature of the maximum of the homogeneous crystal nucleation rate and the conventionally defined glass transition temperature, Tg, coincide widely. This common knowledge is in contrast to Fig.1. I have also serious concerns about the distinction into metastable and unstable states as indicated there.
On the other hand, the main content of the paper consists in the decription of „Technologies for Vitrification based Cryopreservation“. I have not seen such a comprehensive overview on this topic so far. By this reason, I consider the paper as a highly interesting overview despite mentioned above reservations. In further advancing the described spectrum of methods, people for sure will also improve their understanding of the origin of the methods employed and their possible future development.
By these reasons, I recommend publication in the present form.
